# TGF-β ligand cross-subfamily interactions in the response of *Caenorhabditis elegans* to a bacterial pathogen

**Emma Jo Ciccarelli**[1,2], **Zachary Wing**[1], **Moshe Bendelstein**[1], **Ramandeep Kaur Johal**[1], **Gurjot Singh**[1], **Ayelet Monas**[1], **Cathy Savage-Dunn**[1,2]*

**1** Department of Biology, Queens College, City University of New York, New York City, New York, United States of America, **2** PhD Program in Biology, The Graduate Center, City University of New York, New York City, New York, United States of America

* cathy.savagedunn@qc.cuny.edu

**Data Availability Statement:** The authors confirm that all data underlying the findings are fully available without restriction. All relevant data are

## Abstract

The Transforming Growth Factor beta (TGF-β) family consists of numerous secreted peptide growth factors that play significant roles in cell function, tissue patterning, and organismal homeostasis, including wound repair and immunity. Typically studied as homodimers, these ligands have the potential to diversify their functions through ligand interactions that may enhance, repress, or generate novel functions. In the nematode *Caenorhabditis elegans*, there are only five TGF-β ligands, providing an opportunity to dissect ligand interactions in fewer combinations than in vertebrates. As in vertebrates, these ligands can be divided into bone morphogenetic protein (BMP) and TGF-β/Activin subfamilies that predominantly signal through discrete signaling pathways. The BMP subfamily ligand DBL-1 has been well studied for its role in the innate immune response in *C. elegans*. Here we show that all five TGF-β ligands play a role in survival on bacterial pathogens. We also demonstrate that multiple TGF-β ligand pairs act nonredundantly as part of this response. We show that the two BMP-like ligands–DBL-1 and TIG-2–function independently of each other in the immune response, while TIG-2/BMP and the TGF-β/Activin-like ligand TIG-3 function together. Structural modeling supports the potential for TIG-2 and TIG-3 to form heterodimers. Additionally, we identify TIG-2 and TIG-3 as members of a rare subset of TGF-β ligands lacking the conserved cysteine responsible for disulfide linking mature dimers. Finally, we show that canonical DBL-1/BMP receptor and Smad signal transducers function in the response to bacterial pathogens, while components of the DAF-7 TGF-β/Activin signaling pathway do not play a major role in survival. These results demonstrate a novel potential for BMP and TGF-β/Activin subfamily ligands to interact and may provide a mechanism for distinguishing the developmental and homeostatic functions of these ligands from an acute response such as the innate immune response to bacterial pathogens.

within the paper and its Supporting Information files.

**Funding:** This work was funded by the National Institutes of Health, National Institute of General Medical Sciences (R15GM112147 to CSD) and National Institute on Aging (R21AG075315 to CSD, nih.gov). The funders had no role in study design, data collection and analysis, decision to publish, or preparation of the manuscript.

**Competing interests:** The authors have declared that no competing interests exist.

## Author summary

The first line of defense upon exposure to a pathogen consists of innate immunity, which includes barrier functions and cell-cell communication. These cell-cell communication signaling pathways are highly conserved across species, enabling the use of simpler genetically tractable model organisms for their study. One such signaling pathway is the Transforming Growth Factor beta (TGF-β) pathway, which is conserved from invertebrates to vertebrates. TGF-β signaling ligands can be broadly divided into two major groups: the TGF-β/Activin group and the bone morphogenetic protein (BMP) group. There is compelling evidence that heterodimers have biological activities that differ from homodimers, but to this time, heterodimers have only been observed within subfamilies and not across subfamilies. In humans, there are 33 TGF-β ligands. In comparison, there are only five ligands in the nematode *C. elegans*, providing an opportunity to dissect ligand interactions in fewer combinations than in vertebrates. In this work, we show that all five ligands contribute to survival on bacterial pathogens to different extents and with specificity concerning the pathogen. Strikingly, genetic evidence and structural modeling support the existence of cross-subfamily interaction between BMP and TGF-β/Activin ligands.

## Introduction

TGF-β signaling is a highly conserved mechanism that plays significant roles in development in organisms ranging from invertebrates to humans [1–4]. This signaling family also plays an important role in the post-developmental adult in maintaining homeostasis and repair [3,5–7], as well as serving as one of the highly conserved signaling mechanisms involved in the immune response [8–14]. The canonical TGF-β family signaling cascade is initiated by binding of a dimerized ligand to a heterotetrametric receptor complex and mediated through activation of the Smad proteins to regulate transcription [3,15]. The human genome contains 33 TGF-β ligands, seven type I receptors, and five type II receptors [3,16–18]. Currently, active research is focused on the mechanisms that mediate specific or promiscuous interactions between ligands and receptors and the consequences for Smad signaling. These mechanisms have profound implications for normal development and disease [19–25]. The number of potential interactions, however, increases the difficulty of these analyses. The reduced repertoire of ligands and receptors in invertebrate organisms provides an opportunity to study ligand-receptor functional interactions more completely. Due to the conservation of TGF-β signaling pathways, studies in the genetically tractable organisms Drosophila and *C. elegans* have resulted in universal insights into conserved signaling mechanisms.

In *C. elegans*, there are five ligands associated with TGF-β signaling: DBL-1, DAF-7, TIG-2, TIG-3, and UNC-129. DAF-7 and TIG-3 are most appropriately classified as TGF-β/Activin-like ligands, whereas DBL-1 and TIG-2 share the most similarity with BMP ligands in mammals [26,27]. Of the five TGF-β ligands in *C. elegans*, only two have well-characterized signaling pathways through which they signal [27,28]. The TGF-β-like ligand DAF-7 is the regulator of the dauer pathway, and its expression in favorable conditions prevents entry into the dauer larval stage, an alternative third larval (L3) stage specialized for harsh environmental conditions [29,30,31]. The BMP-like ligand DBL-1 activates the BMP-like pathway in the worm and plays a significant role in development [32,33,34]. UNC-129, TIG-2, and TIG-3 have been classified (as BMP-like or TGF-β-like) based on their structural characteristics but have not been fully associated with all members of either the BMP-like or TGF-β-like signaling pathways in *C. elegans* [26,27].

Previous work from our lab and others has demonstrated that the BMP homolog DBL-1 functions in the innate immune response of *C. elegans* to pathogenic bacteria and fungi by inducing the expression of antimicrobial peptide genes in response to pathogen exposure [35–38]. Here we demonstrate that the DBL-1 ligand is only one of four TGF-β family ligands with a significant role in the immune response to two gram-negative bacterial pathogens. We show that in response to either *Serratia marcescens* or *Photorhabdus luminescens*, multiple TGF-β family ligand mutants have a significant reduction in survival compared to control animals. We also demonstrate that two BMP-like ligands, DBL-1 and TIG-2, act independently of each other. In contrast, we have shown a non-additive relationship between TIG-2/BMP and the TGF-β/Activin-like TIG-3, suggesting that these two distinct ligands act together in the immune response. Finally, we have identified signaling components with similar phenotypes to TIG-2 and TIG-3 in response to bacterial pathogen. Our studies thus uncover novel signaling paradigms that may act more broadly to distinguish distinct physiological outcomes regulated by TGF-β ligands.

## Results

### The BMP-like Ligands DBL-1 and TIG-2 are Independently Required for *C. elegans* Survival on Bacterial Pathogen

*C. elegans* has five TGF-β-like ligands: DBL-1, DAF-7, UNC-129, TIG-2, and TIG-3. It is established that loss of DBL-1, one of two BMP-like ligands in *C. elegans*, reduces animal survival on bacterial pathogen [35–36]. In a variety of physiological contexts, heterodimers of BMP ligands are functionally required or outperform homodimers [39–42]. We therefore considered the possibility that DBL-1/BMP functions with the other BMP-like ligand, TIG-2, in the worm. The DBL-1 pathway has been thoroughly studied in response to exposure to the gram-negative bacterium *Serratia marcescens*, where the worm's survival is significantly hindered by gut colonization as a consequence of the pathogen supplanting the normal bacterial food source [36]. To test the role of TIG-2/BMP, we quantified survival of two *tig-2* mutant strains on *S. marcescens* and compared these survival rates to *dbl-1* and N2 (wild type) control. We used *tig-2(ok3336)*, a 500-bp deletion allele, and *tig-2(ok3416)*, an 800-bp deletion allele, and found that both *tig-2* mutant strains resulted in reduced survival against bacterial infection (Fig 1A). Survival rates for both *tig-2* mutant strains were significantly different compared to control. We further tested the effect of mutating *tig-2* on immunity by analyzing the survival of *tig-2* mutant animals on the more virulent *P. luminescens* bacteria. We found that *tig-2* mutants survived significantly worse than control animals (Fig 1B). Interestingly, *tig-2* mutants had an even more pronounced susceptibility to *P. luminescens* infection than *dbl-1* mutants, suggesting a more significant role for TIG-2 in response to *P. luminescens* infection.

Given that both BMP ligands function in the *C. elegans* immune response, we next considered whether they act independently or together by determining the phenotype of double mutants. If these ligands act independently, we expect the double mutant to have a more severe phenotype than the single mutants, reflecting the disruption of two independent pathways. If they act together, such as in a heterodimer, then we expect the double mutant to have the same phenotype as the single mutants due to the failure of individual ligands to provide physiological function. When grown on *P. luminescens*, *tig-2dbl-1* double mutant animals demonstrated a more pronounced survival reduction than the *tig-2* or *dbl-1* single mutant animals (Fig 1C). Our survival analysis shows an additive effect of these two mutations combined. These results indicate that the two BMP-like ligands–DBL-1 and TIG-2–act independently of each other in the response to *P. luminescens*.

At the time of these experiments, no phenotype had been reported for *tig-2* mutants, so we were encouraged to have identified a biological function for this ligand. To rule out a general

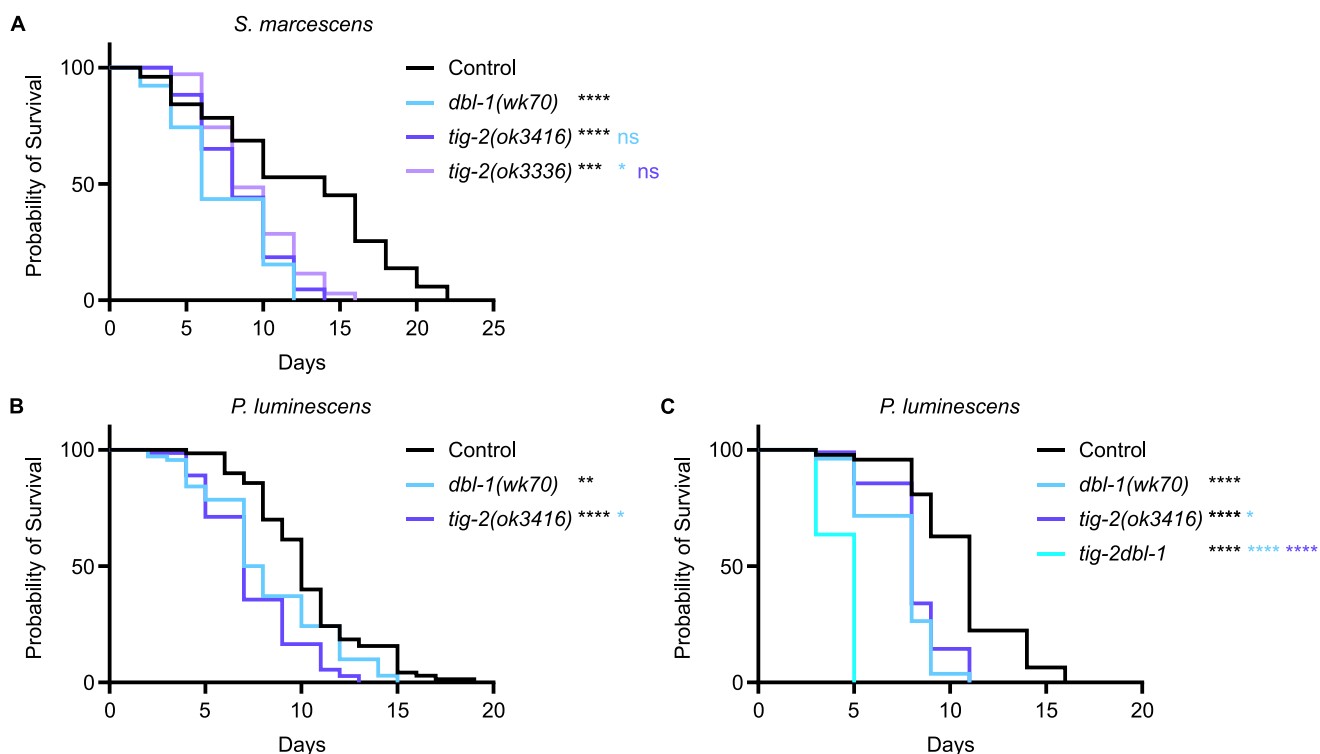

**Fig 1. The BMP-like Ligands DBL-1 and TIG-2 are Required for *C. elegans* Survival on Bacterial Pathogen.** (A) Survival analysis of *dbl-1(wk70)* and two *tig-2* mutants (*tig-2(ok3416)* and *(tig-2(ok3336)*) on *S. marcescens* bacteria. *n* values: Control (51), *dbl-1* (39), *tig-2 (ok3416)* (43), *tig-2 (ok3336)* (35). (B) Survival analysis of *dbl-1(wk70)* and *tig-2(ok3416)* on *P. luminescens* bacteria. *n* values: Control (70), *dbl-1* (70), *tig-2* (73). (C) Survival analysis of *tig-2dbl-1* double mutant on *P. luminescens*. *n* values: Control (94), *dbl-1* (53), *tig-2* (97), *tig-2dbl-1* (55). Statistical analysis done using Log-rank (Mantel-Cox) test. ns p > 0.05; * p ≤ 0.05; ** p ≤ 0.01; **** p < 0.0001. Black asterisks denote significance relative to control, light blue is significance relative to *dbl-1*, and blue-violet is significance relative to *tig-2*.

effect on lifespan, we followed our survival analysis with a lifespan on control *E. coli* bacteria. Both *tig-2* mutants displayed a lifespan similar to control animals (S1 Fig). As expected, animals mutant for *dbl-1* also had an unaffected lifespan on control bacteria. These results indicate that the reduced survival of *tig-2* on bacterial pathogen is specific to its susceptibility to infection and not reflective of a lifespan phenotype. Together these results demonstrate significant roles for the two BMP-like ligands–DBL-1 and TIG-2 –in the response to pathogenic bacterial infection, although independent of each other.

## Five TGF-β Ligands Demonstrate Involvement in Survival Against Bacterial Infection

We next were interested in determining if the remaining TGF-β ligands–DAF-7, UNC-129, and TIG-3 –play a role in the *C. elegans* immune response. UNC-129 and TIG-3 are "orphan" ligands without known receptors and signaling components. DAF-7 is well characterized as the ligand in the dauer pathway, and *daf-7* mutants demonstrate a high incidence of dauer arrest when grown at 20˚C. For this reason, the strains analyzed alongside *daf-7* were all grown past the dauer/L3 stage to the L4 stage at 15˚C and shifted to 20˚C when moved to pathogen plates.

We found that in mutants of any of the five TGF-β ligands, survival against pathogenic *P. luminescens* infection is significantly reduced, with variability between trials for some ligands (Fig 2A and 2B). In particular, *unc-129* mutants have variable survival outcomes. Surprisingly,

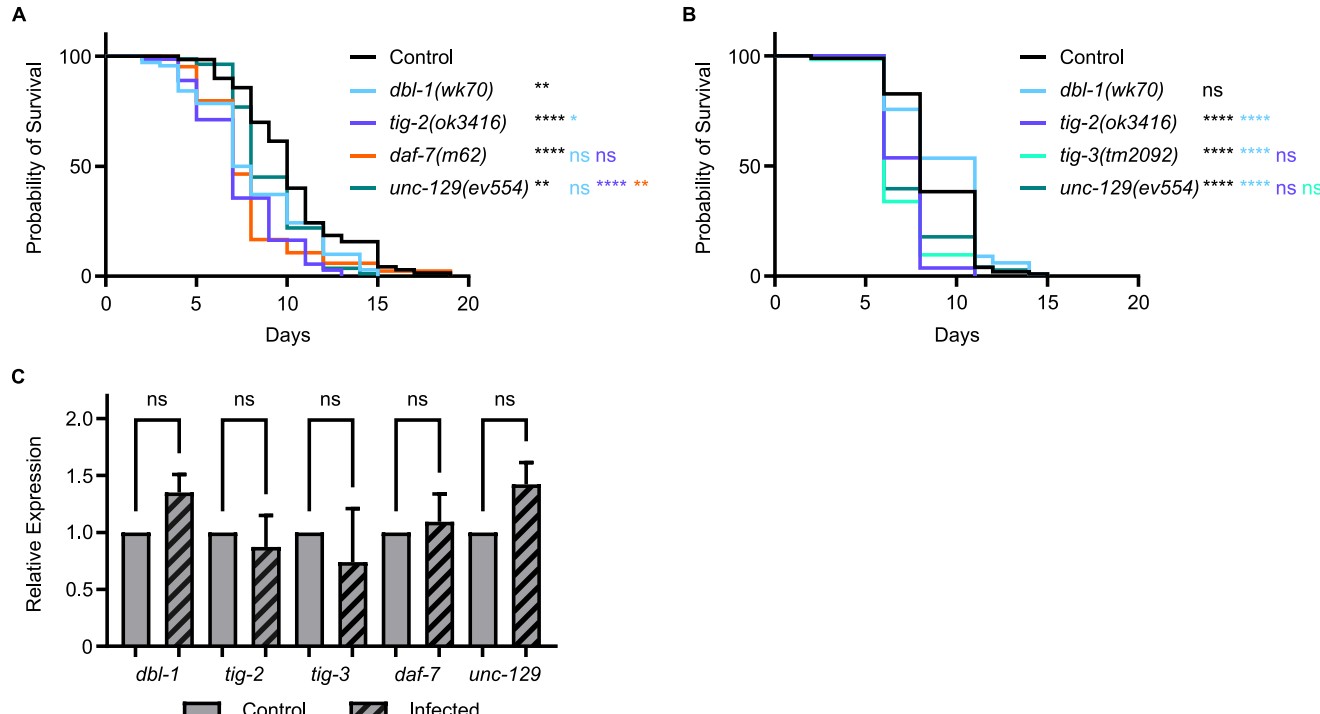

**Fig 2. Five TGF-β Ligands Demonstrate Involvement in Survival Against Bacterial Infection.** (A) Survival of TGF-β ligand mutants on *P. luminescens* bacteria. For this trial, strains were grown at 15˚C to avoid dauer formation by *daf-7* mutants and shifted to 20˚C at L4 when exposed to pathogen. *n* values: Control (70), *dbl-1* (70), *tig-2* (73), *daf-7* (84), *unc-129* (82). (B) Survival of TGF-β ligand mutants on *P. luminescens* bacteria. *n* values: Control (99), *dbl-1* (99), *tig-2* (108), *tig-3* (62), *unc-129* (73). (C) qRT-PCR analysis showing relative expression of TGF-β ligand genes upon 24-hour exposure to *P. luminescens* in control animals. qRT-PCR data represents repeated analyses of two biological replicates. Statistical analysis done using One-way ANOVA with multiple comparisons test. For survivals, statistical analysis done using Log-rank (Mantel-Cox) test. ns p > 0.05; * p ≤ 0.05; ** p ≤ 0.01; **** p < 0.0001. Black asterisks denote significance relative to control, light blue is significance relative to *dbl-1*, blue-violet is significance relative to *tig-2*, orange is significance relative to *daf-7*, and green-cyan is significance relative to *tig-3*.

*dbl-1* mutant animals also have a variable survival pattern on *P. luminescens* despite their well-characterized susceptibility to a variety of bacterial and fungal pathogens. In contrast, TIG-2, TIG-3, and DAF-7 show consistently significant susceptibility in response to *P. luminescens*, concordant with the hypothesis that distinct signaling mechanisms are responsible for the specific responses against different bacterial pathogens.

To determine whether expression of any of these ligands is induced in response to pathogen, we used qRT-PCR to evaluate expression levels of genes encoding TGF-β ligands in wild-type animals. Upon 24-hour exposure to *P. luminescens* bacteria, the relative expression of all five TGF-β ligand genes demonstrated no significant alteration compared to control conditions (Fig 2C). Additionally, analysis of a *dbl-1* transcriptional reporter showed no change in fluorescence levels on pathogen compared to control bacteria (S2 Fig). We conclude that although these ligands function in response to pathogen, they are not subject to widespread transcriptional induction.

## Multiple TGF-β Ligand Pairs Have Nonredundant Roles in the *C. elegans* Response to Bacterial Infection

We next looked for interactions between the remaining *C. elegans* TGF-β ligands upon exposure to bacterial infection by testing double mutants as described above for *tig-2dbl-1* (Fig 1C).

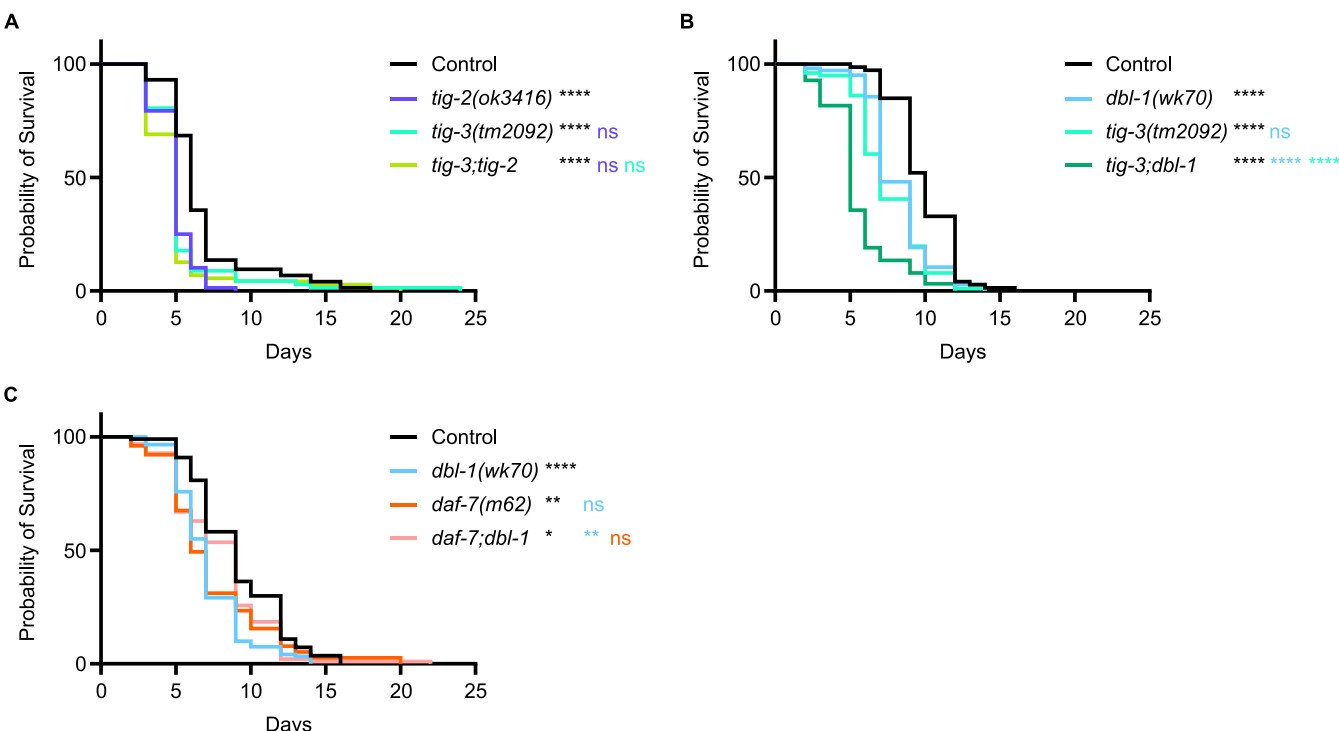

**Fig 3. Two Pairs of TGF-β Ligands Demonstrate Genetic Interactions in the *C. elegans* Response to Bacterial Infection.** (A) Survival analysis of *tig-3 (tm2092);tig-2(ok3416)* double mutants on *P. luminescens* bacteria. *n* values: Control (73), *tig-2* (68), *tig-3* (67), *tig-3;tig-2* (71). (B) Survival analysis of *tig-3 (tm2092);dbl-1(wk70)* double mutants on *P. luminescens* bacteria. *n* values: Control (73), *dbl-1* (104), *tig-3* (101), *tig-3;dbl-1* (126). (C) Survival analysis of *daf-7 (m62);dbl-1(wk70)* double mutants on *P. luminescens* bacteria. *n* values: Control (107), *dbl-1* (117), *daf-7* (75), *daf-7;dbl-1* (92). For this trial, strains were grown at 15°C to avoid dauer formation by *daf-7* mutants and shifted to 20°C at L4 when exposed to pathogen. Statistical analysis done using Log-rank (Mantel-Cox) test. ns p > 0.05; **** p < 0.0001. Black asterisks show significance relative to control, light blue is significance relative to *dbl-1*, blue-violet is significance relative to *tig-2*, green-cyan is significance relative to *tig-3*, and orange is significance relative to *daf-7*.

We analyzed the *tig-3;tig-2* double mutant, deficient in both BMP-like TIG-2 and TGF-β/Activin-like TIG-3, for survival on *P. luminescens*. We found that when grown on bacterial pathogen, *tig-3;tig-2* double mutants demonstrated a survival pattern not significantly different from either *tig-2* or *tig-3* single mutants (Fig 3A). This result contrasts with our analysis of the BMP-like ligands DBL-1 and TIG-2, suggesting that TIG-2/BMP and TIG-3/(TGF-β/Activin) act together in the immune response. If TIG-2 acts independently of DBL-1 but together with TIG-3, we would expect that DBL-1 and TIG-3 also act independently of one another. To test this prediction, we analyzed the survival of *tig-3;dbl-1* double mutants. Survival of the double mutants was significantly reduced compared with either single mutant (Fig 3B). This result is consistent with TIG-3 acting independently of DBL-1.

We also assessed the survival pattern of *daf-7;dbl-1* double mutants on pathogenic bacteria. For this experiment, animals were grown at 15°C to minimize dauer entry in *daf-7* mutants. Survival of *daf-7;dbl-1* double mutants was not worse than that of *dbl-1* and *daf-7* single mutants (Fig 3C), suggesting that these ligands interact rather than acting independently. Surprisingly, the *daf-7;dbl-1* double mutants had significantly better survival than *dbl-1* single mutant animals, which could be explained by a partially antagonistic interaction. This interaction is intriguing as the BMP-like ligand DBL-1 is known to signal through a different Type I receptor and different Smads than the TGF-β/Activin-like ligand DAF-7.

Finally, we compared the survival pattern of double and triple mutants for *unc-129*. Although *unc-129* mutants have weak and variable effects on survival, the *unc-129* gene did

show some potential interactions with other ligand genes. Survival of *unc-129;tig-2* double mutants are not significantly different from *tig-2* single mutants (S3A Fig). In contrast, in *tig-3;unc-129* double mutants, the decreased survival of *tig-3* mutants is suppressed (S3B Fig), suggesting a potential antagonistic interaction between UNC-129 and TIG-3. Overall, our results from these survival analyses indicate a complex relationship between the TGF-β family ligands in the immune response. Most surprisingly, ligands that demonstrate nonredundant relationships are those from different classes of TGF-β ligands–BMP-like TIG-2 with TGF-β/Activin-like TIG-3; and possibly BMP-like DBL-1 with TGF-β/Activin-like DAF-7.

## Canonical BMP Signaling Components are Involved in the *C. elegans* Immune Response

The two TGF-β pathways in *C. elegans* are known to signal through canonical mechanisms upon activation. DBL-1 signals through the BMP-like pathway with ligand binding to the single Type II receptor DAF-4 and the Type I receptor SMA-6. The BMP pathway Smads–SMA-2, SMA-3, and SMA-4 –transduce the intracellular signal allowing for gene regulation. The TGF-β/Activin-like ligand DAF-7 signals through the TGF-β/Activin-like pathway components, including the Type I receptor DAF-1 and the Smads DAF-8, DAF-14, and DAF-3 [32,43–46]. Both pathways converge on the single Type II receptor DAF-4. Previous work has shown that in response to fungal infection of the hypodermis, DBL-1 signals in a pathway utilizing only one receptor-regulated Smad (R-Smad SMA-3) without its typical partner R-Smad SMA-2 or the Co-Smad SMA-4 [37]. Signaling without a Co-Smad is considered non-canonical. We were therefore interested in which components of the BMP-like and TGF-β/Activin-like pathways are required during bacterial infection and whether non-canonical mechanisms are invoked.

Our survival analysis of DBL-1 pathway components shows a consistently reduced survival pattern for R-Smad mutants *sma-2* (Fig 4A) and *sma-3* as well as for Co-Smad mutant *sma-4* (Fig 4B). This result is consistent with a significant role for the BMP pathway Smads in the *C. elegans* response to bacterial pathogen, unlike the results previously shown on fungal infection. Survival analysis of BMP pathway receptors showed a reduced survival pattern for the BMP Type I receptor mutant *sma-6* (Fig 4C). We observe consistently reduced survival for *sma-6* mutants, with slight variability in the significance of the survival rate across trials.

We analyzed the survival patterns of the DAF-7 pathway mutants grown at 15˚C on *P. luminescens* bacteria and found either no phenotype or a mild survival deficit. In particular, R-Smad mutant *daf-8* demonstrated a slightly reduced rate of survival (Fig 4D), while neither R-Smad mutant *daf-14* (Fig 4E) nor Type I receptor *daf-1* (Fig 4F) had significantly altered survival compared to control animals. Thus, the TGF-β/Activin components do not play a major role in pathogen survival, although they may play a modulatory, redundant role in conjunction with BMP signaling components. The surprising implication of this finding is that TGF-β/Activin-like ligands DAF-7 and TIG-3 may interact with BMP signaling components in their roles in the immune response against bacterial pathogen.

## *tig-2* and *tig-3* Mutants Share Reduced Survival and Pumping Rate Phenotypes with *sma-3* Mutants

Thus far, we have shown that TIG-2 (BMP) and TIG-3 (TGF-β/Activin) play more significant roles in survival on *P. luminescens* bacteria than DBL-1/BMP, and that BMP signaling components play a more significant role than TGF-β/Activin signaling components. These observations suggest a model in which TIG-2 and TIG-3 signal through components previously presumed to respond to DBL-1. To test this model, we compared *tig-2* and *tig-3* phenotypes

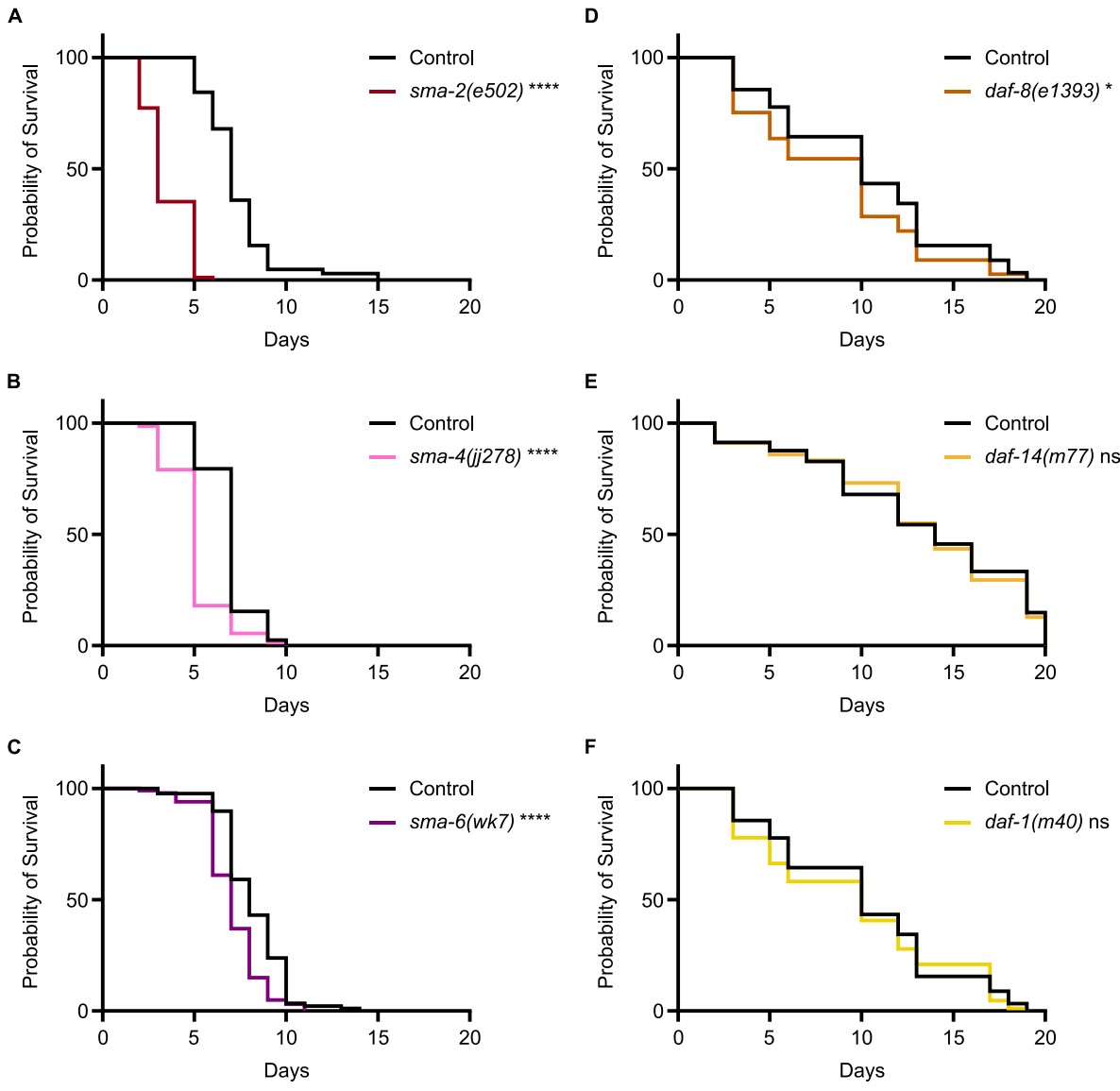

**Fig 4. Canonical BMP Signaling Components are Involved in the *C. elegans* Immune Response.** (A) Survival of DBL-1 R-Smad mutant *sma-2(e502)* on *P. luminescens* bacteria. *n* values: Control (103), *sma-2* (88). (B) Survival of DBL-1 Co-Smad mutant *sma-4(jj278)* on *P. luminescens* bacteria. *n* values: Control (78), *sma-4* (72). (C) Survival analysis of Type I receptor *sma-6(wk7)* on *P. luminescens* bacteria. *n* values: Control (88), *sma-6* (100). (D) Survival of DAF-7 pathway R-Smad mutant *daf-8(e1393)* on *P. luminescens* bacteria. *n* values: Control (90), *daf-8* (77). (E) Survival of DAF-7 pathway R-Smad mutant *daf-14(m77)* on *P. luminescens* bacteria. *n* values: Control (81), *daf-14* (78). (F) Survival analysis of Type I receptor *daf-1(m40)* on *P. luminescens* bacteria. *n* values: Control (90), *daf-1* (86). Statistical analysis done using Log-rank (Mantel-Cox) test. ns p > 0.05; * p ≤ 0.05; **** p < 0.0001.

with those of *sma-3* and *dbl-1*. Survival analysis directly comparing *dbl-1* and *sma-3* mutants on *P. luminescens* demonstrates that while both strains have a decreased survival pattern as compared to control, *sma-3* mutants have a significantly reduced survival rate compared to *dbl-1* mutants (Fig 5A), suggesting that SMA-3 is responding to other signaling ligands instead of or in addition to DBL-1. Notably, *tig-2* and *tig-3* survival phenotypes are highly similar to those of *sma-3* mutants (Fig 5B), consistent with a model in which TIG-2 and TIG-3 signal through R-Smad SMA-3 in response to *P. luminescens* pathogen.

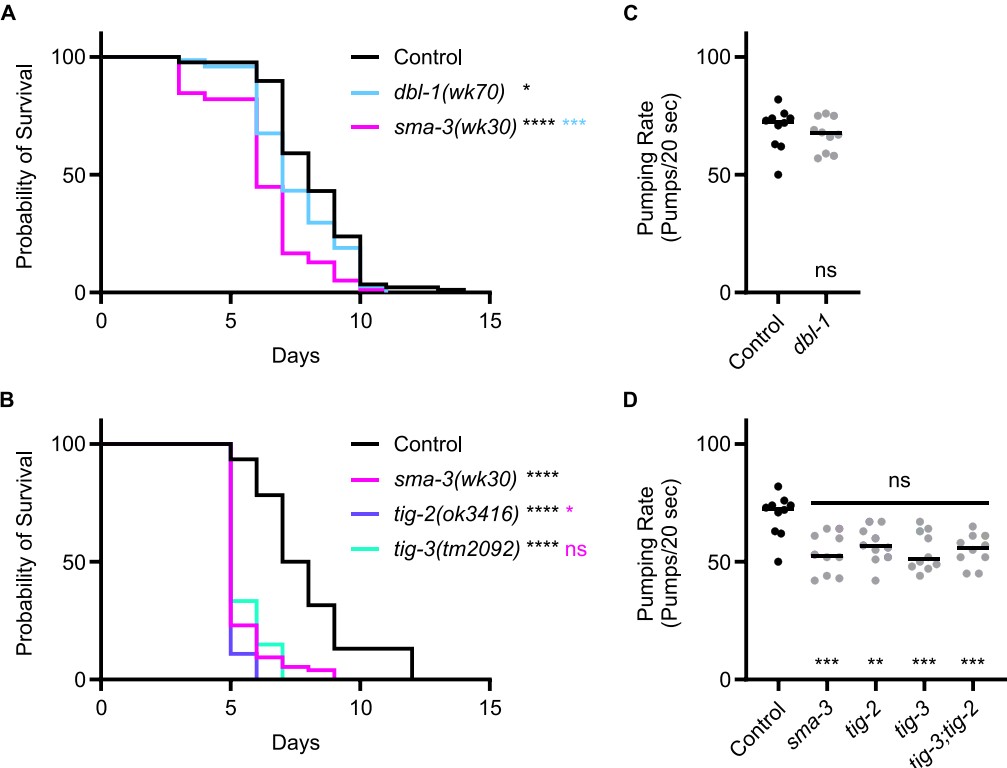

**Fig 5. *tig-2* and *tig-3* Mutants Share Reduced Survival and Pumping Rate Phenotypes with *sma-3* Mutants.** (A) Survival of *dbl-1(wk70)* and *sma-3(wk30)* on *P. luminescens* bacteria. *n* values: Control (88), *dbl-1* (74), *sma-3* (78). Black asterisks are compared to control. Light blue asterisks are compared to *dbl-1*. (B) Survival of *sma-3(wk30)* mutants compared to *tig-2 (ok3416)* and *tig-3(tm2092)* on *P. luminescens* bacteria. Black asterisks are compared to control. Magenta asterisks are compared to *sma-3*. *n* values: Control (92), *sma-3* (74), *tig-2* (55), *tig-3* (54). Statistical analysis for all survivals done using Log-rank (Mantel-Cox) test. Survivals were repeated. (C) Pumping rate per 20 seconds for *dbl-1(wk70)* compared to control. Statistical analysis done using t test. (D) Pumping rate per 20 seconds for *sma-3(wk30)*, *tig-2(ok3416)*, *tig-3 (tm2092)*, and *tig-3;tig-2*. Black asterisks are compared to control. Magenta asterisks are compared to *sma-3*. Statistical analysis done using One-way ANOVA with multiple comparisons test. *n* values for all pumping rate experiments: ten worms per strain. Pumping rate experiments were repeated on independent biological samples. ns p > 0.05; * p ≤ 0.05; ** p ≤ 0.01; *** p ≤ 0.001; **** p < 0.0001.

To test this model further, we analyzed an additional phenotype associated with response to *P. luminescens*, reduced pharyngeal pumping rate (Ciccarelli *et al*., 2023 [35]). Pharyngeal pumping rate analysis revealed *sma-3*, but not *dbl-1*, mutants have a significant reduction in pharyngeal pumping rate, and this reduction is indistinguishable from that of *tig-2* and *tig-3* mutants, as well as of the *tig-3;tig-2* double mutant (Fig 5C and 5D). This similarity further strengthens the model that TIG-2 and TIG-3 signal through SMA-3 in this context.

## Structural Modeling of Potential Protein-Protein Interaction Between TIG-2/BMP and TIG-3/(TGF-β/Activin)

The nonredundant action observed between TIG-2/BMP and TIG-3/(TGF-β/Activin) in the *C. elegans* immune response could be evidence of TIG-2/TIG-3 heterodimers. Alternatively, TIG-2 and TIG-3 homodimers could act together through a receptor clustering mechanism [47]. To determine whether heterodimers of TIG-2 and TIG-3 are feasible, we performed structural modeling *in silico*. Using the ColabFold [48] implementation of AlphaFold2 [49], we

asked whether confident structural predictions could be made for TIG-2 homodimers, TIG-3 homodimers, and TIG-2/TIG-3 heterodimers. We first analyzed interactions between mature bioactive domains using consensus proteolytic cleavage sites (see Materials and Methods; S4 Fig). Using the mature domain of TIG-2, we generated a strongly supported model for the TIG-2 homodimer that recapitulates the well-known "butterfly" structure of TGF-β family ligand dimers (Fig 6). In contrast, the potential for the TIG-3 mature bioactive domain to form homodimers was not supported by ColabFold, with a poor interface predicted template modeling (ipTM) score of 0.288. Modeling TIG-2 and TIG-3 together, however, resulted in a heterodimer with a high ipTM score of 0.783, similar to that of the TIG-2 homodimer (0.875), as well as to those of known heterodimers (S5 Fig). Intriguingly, both TIG-2 and TIG-3 lack the conserved cysteine residue that mediates interchain disulfide bridges in most TGF-β family ligands (S6 Fig). Two other TGF-β family members also lacking dimerization cysteines are BMP15 and GDF9 [50]. These ligands form stable homo- and heterodimers without interchain disulfide bridges. In fact, GDF9/BMP15 heterodimers, known as cumulin, can form from homodimers by subunit exchange [50].

In vivo, prodomains present in the uncleaved precursors have been shown to be required for heterodimer formation between BMP4 and BMP7 [51]. We, therefore, performed structural modeling with pro-proteins (prodomain + mature domain). These procomplex models indicate additional potential contacts between pro-TIG-2 and pro-TIG-3 (Fig 7). The TIG-2/TIG-2 procomplex resembles a BMP dimer in an open-arm conformation, with each monomer prodomain remaining separate but making contact with the mature domains (Fig 7A and 7C). The predicted TIG-2/TIG-3 procomplex exhibits an asymmetric conformation, with the TIG-3 prodomain crossing over to interact with the TIG-2 pro- and mature domains (Fig 7B and 7C). Thus, structural modeling of the pro-proteins also supports the possibility of heterodimer formation consistent with our functional genetic analysis. This model can now be tested with biochemical analysis of protein-protein interactions.

## Discussion

The multiplicity of TGF-β family ligands, receptors, and Smad signal transducers may support the generation of diverse context-dependent outcomes through combinatorial interactions. Biochemical and computational approaches are useful in identifying the principles mediating the potential protein-protein interactions between these components, but these approaches must be complemented by functional in vivo analyses. The nematode *C. elegans* has five TGF-β ligands and two characterized TGF-β family signaling pathways [27,28], providing a powerful in vivo system in which to study TGF-β ligand functions in the context of an intact organism. Critically, TGF-β signaling mechanisms are conserved in this organism. Here we demonstrate that all five TGF-β ligands play a role in survival on bacterial pathogen. Furthermore, two ligand pairs consisting of a BMP subfamily and a TGF-β/Activin subfamily member (TIG-2/TIG-3 and DBL-1/DAF-7) show evidence of interacting in the response to gram-negative bacterium *P. luminescens*.

These studies were initiated with the well-characterized BMP-like DBL-1 signaling pathway, which is associated with significant roles in development and body size regulation but has also been shown to play an important role in the immune response against both bacterial and fungal pathogens [36,37,53]. Comparing the response to the gram-negative bacteria *S. marcescens* and *P. luminescens*, we obtained evidence implicating a level of specificity of ligand responses for particular bacteria. Although both BMP-like ligands, DBL-1 and TIG-2, play a significant role in the response against *S. marcescens*, TIG-2 has a more significant role than DBL-1 in the response against *P. luminescens*. These results suggest that the function of DBL-1

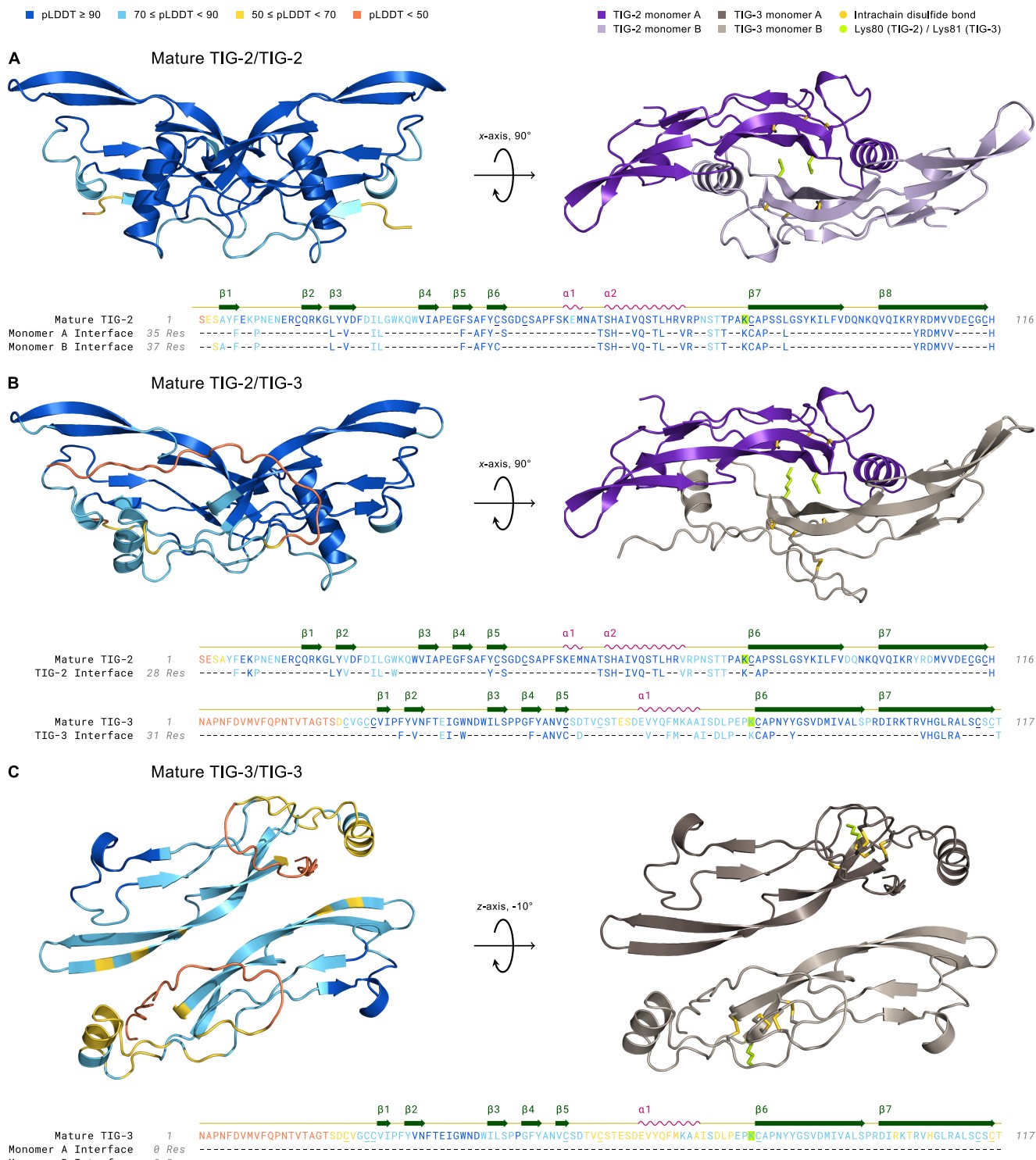

**Fig 6. AlphaFold2-Multimer Structural Modeling of TIG-2 and TIG-3 Mature Ligand Homo- and Heterodimer Complexes.** Left complex: 3D structures are colored by pLDDT confidence score with deep blue corresponding to regions of very high confidence and orange representing regions of very low confidence or disorder. Right complex: The mature dimer as in the left panel but colored by monomer and following geometric rotation. All cysteine residues (underlined) in mature TIG-2 (with 15–81, 44–113, 48–115 disulfides) and TIG-3 (with 22–26, 25–82, 54–114, 59–116 disulfides) homo- and heterodimers are paired in intrachain disulfide bonds (gold) within the cystine knot domain. The cysteine residue that forms the interchain disulfide bond in the mature dimer is absent in TIG-2 and TIG-3, similar to GDF3, GDF9, and BMP15 [52]. Interestingly, lysine (yellow-green), with its reactive ε-amino group upon deprotonation,

is instead substituted for this cysteine residue in both TIG-2 (Lys80) and TIG-3 (Lys81). Amino Acid Sequence: Residues are colored by pLDDT score. The monomer interface tracks consist of confidently predicted dimer-interface contacts (see Methods). (A) Mature TIG-2(purple)/TIG-2(lavender) exhibits high confidence with respect to per-residue structural modeling (pLDDT: 92.7), pairwise residue alignment confidence (pTM: 0.868), and dimeric interaction confidence (ipTM: 0.875). (B) Mature TIG-2(purple)/TIG-3(gray-beige) is confidently modeled regarding its structure and potential for dimerization (pLDDT: 84.9, pTM: 0.798, ipTM: 0.783). (C) Mature TIG-3(brown)/TIG-3(gray-beige) falls slightly below (pLDDT: 68.8) the threshold for a confident structure prediction (pLDDT ≥ 70), and its predicted ability to homodimerize is very low (ipTM: 0.288). Additionally, no confidently predicted (pTM: 0.522) interface residues exist between TIG-3 monomers.

in the immune response is not a general one-size-fits-all response to bacterial pathogen but rather a more nuanced response with some level of pathogen specificity. Furthermore, we show that DBL-1/BMP and TIG-2/BMP have additive roles in the response to *P. luminescens*, suggesting that they trigger independent responses.

The orphan ligand UNC-129 shows moderate and variably significant changes in survival on *P. luminescens*, suggesting a minor or modulatory role. The TGF-β/Activin-like ligands DAF-7 and TIG-3 demonstrated consistently significant reduced survival patterns on *P. luminescens*, suggesting involvement of these ligands in the response to this bacterial pathogen. qRT-PCR analysis of TGF-β ligand expression shows that with 24-hour exposure to bacterial infection, expression levels of the ligands do not significantly change from control conditions. A *dbl-1p*::GFP transcriptional reporter validates these results for *dbl-1*. Taken together, we conclude that all five of the TGF-β ligands play a role in the immune response, but not at the level of transcriptional induction.

Analysis of the survival patterns of double mutant animals produced results indicating interactions between multiple TGF-β family ligands. Interestingly, the interactions between pairs of BMP-like and TGF-β/Activin-like ligands suggest interaction in a common pathway. Specifically, the *tig-3;tig-2* double mutant survival pattern is no more severe than either single mutant alone. Similarly, the *daf-7;dbl-1* double mutant survival is indistinguishable from that of *daf-7* single mutants. Of these pairs, TIG-2 and TIG-3 play a more major role in the response to *P. luminescens*. We, therefore, used computational modeling to determine whether these ligands could feasibly function as a heterodimer. Using ColabFold, we demonstrated that TIG-2/TIG-3 heterodimers are better supported than TIG-3 homodimers. Furthermore, *tig-2* and *tig-3* expression overlaps in neurons located adjacent to the pharynx, such as AFD and M2. We have shown that SMA-3/Smad is required in pharyngeal muscle for survival on bacterial pathogens [35], so TIG-2 and TIG-3 are produced in a location appropriate for signaling to SMA-3/Smad in the pharynx.

We also identify the canonical BMP-like pathway as required for response to *P. luminescens*, including the Type I receptor (SMA-6), the R-Smads (SMA-2 and SMA-3), and Co-Smad (SMA-4). In contrast, TGF-β/Activin pathway components have minimal effects. We previously provided evidence that DBL-1 acts through SMA-3 to regulate antimicrobial peptide genes *abf-2* and *cnc-2* [35]. The survival deficit of *sma-3* mutants, however, is more similar to that of *tig-2* and *tig-3* mutants. Furthermore, *tig-2*, *tig-3*, and *sma-3* mutants share a pharyngeal pumping defect not seen in *dbl-1* mutants. We have not determined why the *sma-3* survival defect is not worse than that of *tig-2* or *tig-3* mutants, as the *tig-2dbl-1* and *tig-3;dbl-1* phenotypes are. It is possible that activation of SMA-2 in a *sma-3* mutant background partially compensates for the loss of *sma-3* and that this activation is lacking in *tig-2dbl-1* and *tig-3;dbl-1* double mutants, causing a more severe phenotype. The phenotypic similarities provide evidence that TIG-2 and TIG-3 signal through the canonical BMP signaling pathway consisting of SMA-6 Type I receptor and SMA-2, SMA-3, and SMA-4 Smads in the context of bacterial pathogen survival. There is precedence for TGF-β/Activin ligands signaling through BMP-like signaling pathways in vertebrates. For example, in endothelial cells, TGF-β can phosphorylate

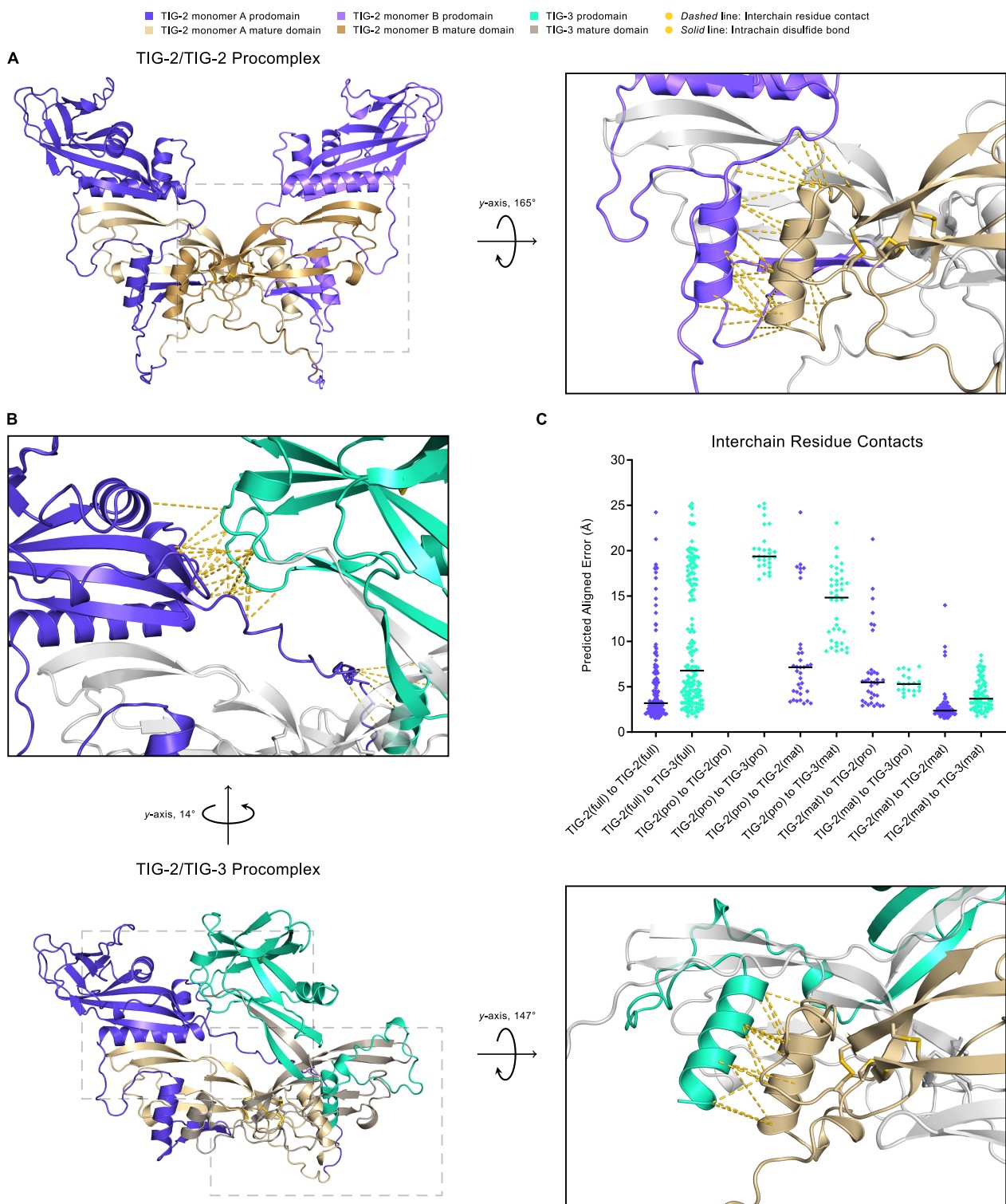

**Fig 7. AlphaFold2-Multimer Structural Modeling of TIG-2/TIG-2 and TIG-2/TIG-3 Procomplexes.** (A) The TIG-2/TIG-2 procomplex exhibits a symmetric open-arm conformation with monomer A in blue-violet (prodomain) and khaki (mature domain) and monomer B in light blue-violet (prodomain) and dark khaki (mature domain). The right panel is the magnified and rotated area defined by the gray dashed box in the left panel. Interchain residue contacts between the mature domain (khaki) of pro-TIG-2 monomer A to the prodomain (light blue-violet) of pro-TIG-2 monomer B are shown as gold *dashed* lines. As is true for the mature form, the TIG-3 homodimer procomplex (not shown) has significantly lower multimer metrics

(pLDDT: 66.5, pTM: 0.354, ipTM: 0.281) and thus a reduced predicted likelihood to homodimerize. (B) The TIG-2/TIG-3 procomplex adopts an asymmetric conformation, with pro-TIG-2 forming an open-arm conformation and pro-TIG-3 (prodomain in green-cyan, mature domain in gray-beige) presenting a crossed-arm conformation. Pro-TIG-2 contains no prodomain cysteines (S4A Fig) and hence is not prodomain disulfide linked to pro-TIG-3. The only two cysteines in the TIG-3 prodomain (S4A Fig) are paired in an intra-prodomain 60–70 disulfide bond (*solid* gold line). The enlarged upper panel corresponds to the upper-left dashed box following rotation to emphasize the interchain residue contacts resulting from the prodomain of pro-TIG-3 (green-cyan) crossing over to interact with the prodomain of pro-TIG-2 (blue-violet). Similarly to A, the right panel corresponds to the lower-right gray dashed box and features interchain residue contacts between the mature domain of pro-TIG-2 (khaki) and the prodomain of pro-TIG-3 (green-cyan). (C) A comparison of interchain residue contacts in TIG-2/TIG-2 (blue-violet diamonds) and TIG-2/TIG-3 (green-cyan diamonds) procomplexes between combinatorial interacting monomer regions (full: full-length domain, pro: prodomain, mat: mature domain) plotted with their predicted aligned error value and column mean (black horizontal line). For example, the column TIG-2(full) to TIG-3(full) contains all interchain residue contacts between pro-TIG-2 and pro-TIG-3. Correspondingly, the column TIG-2(mat) to TIG-3(pro) includes all interchain residue contacts between the mature domain of pro-TIG-2 to the prodomain of pro-TIG-3. The TIG-2/TIG-3 procomplex compares similarly to the TIG-2/TIG-2 procomplex with respect to full-length to full-length (columns 1 and 2), mature domain to prodomain (columns 7 and 8), and mature domain to mature domain contacts (columns 9 and 10).

presumptive BMP Smads Smad1/5 through interaction with type I receptor ALK1 rather than ALK5 [54]. Furthermore, the BMP receptor ALK2 (ACVR1) can be activated by Activin to phosphorylate Smad1/5, and this activity is increased by pathogenic FOP mutations [55]. However, joint action between a BMP and TGF-β/Activin ligand, including potential cross-subfamily heterodimers, has not yet been described in vertebrates to our knowledge, and warrants further consideration.

Interestingly, nonredundant roles for TIG-2, TIG-3, and UNC-129 have recently been identified in neuronal guidance in *C. elegans* [56]. In neuronal guidance, TIG-3/UNC-129 heterodimers are implicated, with TIG-2 being released from a different tissue. In this context, the ligands depend on the BMP signaling components SMA-6 Type I Receptor and SMA-2, SMA-3, and SMA-4 Smads. Relatedly, three Activin ligands are required nonredundantly in Drosophila to regulate photoreceptor identity through the canonical Activin signaling components [57]. In addition to heterodimers, nonredundant functions of TGF-β ligands can be explained by a receptor clustering mechanism that generates higher-order signaling complexes [47]. Our system provides a platform for studying the functional interactions between multiple TGF-β ligands. Interaction between BMP and TGFβ/Activin subfamily members may provide a mechanism for distinguishing developmental and homeostatic ligand functions from an acute response such as the innate immune response to bacterial pathogens.

## Materials and methods

### Nematode strains and growth conditions

*C. elegans* were maintained on *E. coli* (DA837) using EZ worm plates containing streptomycin. Worms were maintained at 20°C, except in experiments using dauer pathway mutants. These strains were maintained at 15°C to prevent entry into dauer. The N2 strain is used as a control. All strains used in this study are: N2, *daf-1(m40)*, *daf-7(m62)*, *daf-8(e1398)*, *daf-14(m77)*, *sma-2(e502)*, *sma-3(wk30)*, *sma-4(jj278)*, *sma-6(wk7)*, *tig-2(ok3336)*, *tig-2(ok3416)*, *tig-3(tm2092)*, *unc-129(ev554)*, *daf-7(m62);dbl-1(wk70)*, *tig-2(ok3416)dbl-1(wk70)*, *unc-129(ev554);tig-2 (ok3416)*, *tig-3(tm2092);tig-2(ok3416)*, *tig-3(tm2092);unc-129(ev554)*, *tig-3(tm2092);unc-129 (ev554);tig-2 (ok3416)*, *dbl-1p*::GFP. Genetic data were obtained from WormBase [58].

### Bacteria

Control bacteria in all experiments is *E. coli* strain DA837, cultured at 37°C. *S. marcescens* strain Db11 (cultured at 37°C) and *P. luminescens* (cultured at 30°C) were used for bacterial pathogen in survival analyses. *S. marcescens* (Db11) is seeded on EZ worm plates containing

streptomycin and grown overnight at 37˚C. *P. luminescens* is seeded on EZ worm plates with no antibiotic and grown overnight at 30˚C.

## Survival analysis

Survival plates were prepared at least one day prior to use. Each plate was seeded with 500μl pathogenic bacteria in a full lawn. FuDR, at a concentration of 50 μM per plate, was used to prevent reproduction and reduce the incidence of matricide during survival analysis. Survivals were conducted at 20˚C, except survivals using dauer pathway mutants, which were conducted at 15˚C to minimize dauer entry (unless otherwise stated). Survival analyses were repeated for all ligand and DBL-1 pathway experiments. All graphs made using GraphPad Prism and statistical analysis performed using Log-rank (Mantel-Cox) test.

## Fluorescence imaging

Fluorescence imaging of the *dbl-1p*::GFP transcriptional reporter strain was done using a Zeiss ApoTome with AxioVision software and a (20X) objective. Exposure times were kept consistent. Image analysis and fluorescence intensity measurements were done using ImageJ software. Fluorescence intensity was measured for ten nuclei in each of four worms per bacterial exposure condition.

## qRT-PCR Analysis

qRT-PCR analysis was performed on collected samples of N2 control animals with or without 24-hour *P. luminescens* pathogen exposure. RNA was obtained by a previously described protocol [59] and followed by use of the Qiagen RNeasy miniprep kit (Cat. No. 74104). cDNA was made using SuperScript IV VILO Master Mix (Cat. No.11756050) from Invitrogen, Waltham, MA. qRT-PCR analysis was done using Power SYBR Green PCR Master Mix (Cat. No. 4367659) from Applied Biosystems, Waltham, MA. qRT-PCR was repeated on separate biological replicates. Delta delta Ct analysis was done using Applied Biosystems and StepOne software. Graphs made using GraphPad Prism software.

## Structural modeling

SignalP 6.0 was used to predict the signal sequence for TGF-β family members. No signal sequence for TIG-3 isoform A could be determined, and this potential isoform was not considered further. For TIG-3 isoform B, the prodomain and mature domain are defined using a consensus cleavage site [60]. The TIG-2 cleavage site was determined by manual sequence analysis using the $(R/K)-X_n-(R/K)$ motif as a reference where X is an amino acid sequence of length $n$ such that $n$ is even and $0 \leq n \leq 6$ [52]. The existence of multiple functional cleavage sites is common [52], and two such sites (RSRR and KVKR) were identified. Both cleavage sites are supported by structural modeling and preserve the cystine knot domain. The main distinction between the two sites is the length of the initial disordered region of the mature ligand (RSRR: 134-residue vs. KVKR: 116-residue mature form). Previously reported cleavage sites were used for the remaining TGF-β family members [52], The full-length domain (Fig 7C) is defined as the entire monomer sequence, excluding the signal sequence. Dimeric structures were modeled using the ColabFold (version 1.5.2) implementation of AlphaFold2--multimer (version 3). The AlphaFold2_mmseqs2 notebook was run under the default settings, and amber relaxation was applied to all five structural models. The model assigned rank one was selected for visualization and analysis. Interchain residue contacts (Figs 6 and 7C) within 4 Å and their predicted aligned error (PAE) value were identified using UCSF ChimeraX

(version 1.5). One-dimensional amino acid sequence tracks (Fig 6) include only dimer-interface residues that are within a distance of 4 Å and have a low positional error (PAE < 5 Å). Images of three-dimensional structures were rendered using PyMOL (version 2.5.4). pLDDT color coding and secondary structure annotation of one-dimensional amino acid sequences were generated using iCn3D (version 3.24.2) and reformatted. The alpha helices used in secondary structure labeling are modified vectors from BioRender.com.

## Supporting information

**S1 Fig. Lifespan analysis of *dbl-1(wk70)*, *tig-2(ok3416)*, and *tig-2(ok3336)* on nonpathogenic *E. coli* strain DA837.** *n* values: Control (51), *dbl-1* (41), *tig-2* (56), *tig-2* (51). (PDF)

**S2 Fig. Reporter imaging of *dbl-1p*::GFP transcriptional reporter on control *E.* coli and 24-hour infection on *P. luminescens* bacteria.** No significance. (PDF)

**S3 Fig. Survival analysis of *unc-129(ev554);tig-2(ok3416)*, *tig-3(tm2092);unc-129(ev554)*, and *tig-3(tm2092);unc-129(ev554);tig-2(ok3416)*.** (A) *n* values: Control (73), *tig-2* (68), *unc-129* (70), *unc-129;tig-2* (83). (B) *n* values: Control (73), *tig-3* (67), *unc-129* (70), *tig-3;unc-129* (83). (C) *n* values: Control (73), *tig-2* (68), *tig-3* (67), *unc-129* (70), *tig-3;unc-129;tig-2* (79). Statistical analysis for survival analyses done using Log-rank (Mantel-Cox) test. ns p > 0.05; * p ≤ 0.05; ** p ≤ 0.01; **** p < 0.0001. Black asterisks denote significance relative to control, blue-violet is significance relative to *tig-2*, green-cyan is significance relative to *tig-3*, and teal is significance relative to *unc-129*. (PDF)

**S4 Fig. Structural Modeling Sequence Data.** (A) Annotated TIG-2 and TIG-3 isoform B amino acid sequences. The signal sequence (black) is underlined, and the cleavage site is in boldface. The prodomain (blue-violet in TIG-2, green-cyan in TIG-3) includes the cleavage site, which separates the prodomain from the mature ligand (dark khaki in TIG-2, gray-beige in TIG-3). Cysteine residues participating in intrachain disulfide bonds are highlighted in gold. The lysine residue that replaces the dimerization cysteine is colored yellow-green. (B) Subfamily, species, Uniprot entry, signal sequence, and cleavage site for modeled TGF-β family members. (a) [60] (b) SignalP 6.0. (c) Manual sequence analysis. (d) [52]. (PDF)

**S5 Fig. Benchmarking AlphaFold2-Multimer Structure Predictions against Experimentally Tested TGF-β Family Ligand Interactions.** TIG-2 and TIG-3 structure predictions benchmarked against predictions for experimentally reported dimers [50,61–63] and the non-homodimeric inhibin α subunit [64]. Average pLDDT divided by 100 (navy), pTM (khaki), and ipTM (amber) metrics are grouped for each complex. Lavender and gray dashed lines correspond to TIG-2/TIG-3 and TIG-3/TIG-3 ipTM scores, respectively. Monomers are color-coded according to TGF-β subfamily, with BMP in blue-violet and TGF-β/Activin in green-cyan. Both mature and procomplexes of the TIG-2 homodimer and TIG-2/TIG-3 heterodimer converge with known dimers. In contrast, the mature and pro-forms of the TIG-3 homodimer markedly diverge with inhibin α that does not homodimerize. Mature activin A, the inhibin βA homodimer, groups with mature inhibin α despite the established ability of inhibin βA to homodimerize. Activin A procomplex, however, expectedly groups with the reported dimers. The mature and pro results together support the necessity of prodomain-mediated dimerization for activin A, as previously shown [65]. Reported non-disulfide-linked mature dimers

include GDF-9/GDF-9, GDF-9/BMP-15, and BMP-15/BMP-15. Procomplex multimer metrics are observably lower (except for pTM and iPTM scores for activin A and inhibin B) than the equivalent mature form.
(PDF)

**S6 Fig. Multiple Sequence Alignment using Clustal Omega of Mature TIG-3, Inhibin α, GDF-9, BMP-15, Inhibin βA, TIG-2, BMP-2, and DBL-1.** The conventional dimerization cysteine is highlighted in gold and replaced by lysine (yellow-green) in TIG-2 and TIG-3 and by serine (cyan) in GDF-9 and BMP-15. Consensus symbols (asterisk, colon, period) are defined according to standard Clustal Omega notation.
(PDF)

## Acknowledgments

Some strains were provided by the Caenorhabditis Genetics Center, which is supported by the National Institute of Health–Office of Research Infrastructure Programs (P40 OD010440). We are grateful to the lab of Roger Pocock for sharing double and triple mutants of TGF-β ligand genes. We thank Jan Christian for helpful comments on the manuscript. This work was carried out in partial fulfillment of the requirements for the Ph.D. degree from the Graduate Center of City University of New York (EJC).

## Author Contributions

**Conceptualization:** Emma Jo Ciccarelli, Zachary Wing, Cathy Savage-Dunn.

**Funding acquisition:** Cathy Savage-Dunn.

**Investigation:** Emma Jo Ciccarelli, Zachary Wing, Moshe Bendelstein, Ramandeep Kaur Johal, Gurjot Singh, Ayelet Monas.

**Project administration:** Cathy Savage-Dunn.

**Supervision:** Cathy Savage-Dunn.

**Visualization:** Zachary Wing.

**Writing – original draft:** Emma Jo Ciccarelli, Cathy Savage-Dunn.

**Writing – review & editing:** Emma Jo Ciccarelli, Zachary Wing, Cathy Savage-Dunn.

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
