## [Decision Letter · Decision Letter 0]

10 Oct 2023

Dear Dr Savage-Dunn,

Thank you very much for submitting your Research Article entitled 'TGF-β Ligand Cross-Subfamily Interactions in the Response of Caenorhabditis elegans to Bacterial Pathogens' to PLOS Genetics.

The manuscript was fully evaluated at the editorial level and by independent peer reviewers. The reviewers appreciated the attention to an important problem, but raised some substantial concerns about the current manuscript. Based on the reviews, we will not be able to accept this version of the manuscript, but we would be willing to review a much-revised version. We cannot, of course, promise publication at that time.

If you decide to revise the manuscript for further consideration at PLOS Genetics, please aim to resubmit within the next 60 days, unless it will take extra time to address the concerns of the reviewers, in which case we would appreciate an expected resubmission date by email to plosgenetics@plos.org.

We are sorry that we cannot be more positive about your manuscript at this stage. Please do not hesitate to contact us if you have any concerns or questions.

Yours sincerely,

Danielle A. Garsin

Academic Editor

PLOS Genetics

Gregory P. Copenhaver

Editor-in-Chief

PLOS Genetics

Reviewer's Responses to Questions

**Comments to the Authors:**

Reviewer #1: This is an excellent study from the Savage-Dunn lab investigating the roles of the five TGFB ligands in C. elegans in resistance to bacterial pathogen infection. One ligand was previously known to function in this role Dbl-1. They show, surprisingly, that all five ligands play a role in resistance to bacterial pathogen infection. They show that DBL-1 and TIG-2 BMP ligands function additively to each other in survival to these pathogens, whereas the BMP/Activin pair, TIG-2 and TIG-3, do not function additively to each other and instead each is independently required, suggesting they act as a heterodimer. Interestingly, their structural modeling of TIG-2 and TIG-3 supports formation of TIG-2-TIG-3 heterodimers, TIG-2 homodimers, but poor results were found for TIG-3 homodimers. Furthermore, they provide compelling evidence that TIG-2 (BMP) and TIG-3 (Activin-like) function through the BMP Smad, SMA-3, and not through the Activin Smad or Activin pathway Type I receptor. The only weakness of the paper is not showing direct involvement of the immune response for the TIG-2/TIG-3 response, as the role for DBL-1 is already well known. Mostly minor comments are below.

Line 133, “The larger deletion allele resulted in a survival defect similar to that of dbl-1 mutants and more pronounced than the survival defect caused by the 500 bp deletion.” Since the difference between the two TIG-2 alleles is not significantly different, stating a ‘more pronounced’ survival defect between them would be better omitted.

Line 177-79, it is worth mentioning for the non-worm specialist when the dauer period is relative to L4, so the experimental scheme is more clear.

Fig 2B, it seems that the control also has a variable response to P. luminescens bacteria, when comparing it to Fig 1 panel values. The authors should comment on that. Could the P. luminescens bacteria be changing in any way that changes the survival response? Or other environmental factors that might be playing a role?

Line 222-24, Since the double mutant has the same survival rate as the single mutants, why does that suggest they function in ‘a cooperative manner’ and what is meant by that exactly? Earlier state that it would suggest “they act together” for example as a heterodimer, when making the dbl-1 TIG double mutant. Usually one refers to acting in the same pathway (if no stronger phenotype) or in different pathways (stronger phenotype). Indeed in referring to the daf-7, dbl-1 double mutant on line 230 it is interpreted as a single pathway. Cooperativity is mentioned in that paragraph too, the abstract, and in the Discussion as well, and again is unclear. Cooperative protein interactions mean something completely different, hence, this terminology in this context is a bit confusing.

Fig 3B, shouldn’t the daf-7 mutant alone be shown too for comparison?

Line 242, a typo. One of those should be TIG-3.

Figure 4, might there be some functional overlap or redundancy between the Type I receptors and R-Smads.

Figure 7A, right panel, the “Interchain Residue Contacts” label is located in panel C, better if moved up into panel A region.

Discussion

Line 431: “Here we demonstrate that all five TGF-β ligands play a role in the immune response

against bacterial pathogen”. All 5 ligands have been demonstrated to play a role in survival to pathogenic bacteria, does that necessarily mean they play a role in the immune response? The results presented seem insufficient to state that. Either further evidence should be provided or these statements modified.

Zhang and Zhang, 2012, and Zugasti and Ewbank, 2009, are missing from the Reference list.

Reviewer #2: The authors studied the role of five TGF-β ligands in C. elegans defense against killing by P. luminescens or S. marcescens. The results suggest a co-relationship between cross-subfamily cooperation of the TGF-β ligands. In addition to the DBL-1 pathway, the BMP signaling components SMAD-2/4/6 showed involvement in C. elegans survival against infections. The authors raised a novel idea of the potential interaction between BMP and TGF-β/Activin subfamily ligands. However, the data does not seem to fully support the claims, and additional work seems needed to prove the involvement of the ligands in the overall immune response. The authors might need to study more bacteria to justify their title ‘TGF-β Ligand Cross-Subfamily Interactions in the Response of Caenorhabditis elegans to Bacterial Pathogens.’

1- All five TGF-β ligands play a role in the immune response – the authors demonstrate immune response solely based on survival studies.

2- The immune response is not a general one-size-fits-all response to bacterial pathogens but rather a more nuanced response with some level of pathogen specificity. Why only two-gram negative bacteria were used?

3- The authors looked for interactions between the C. elegans TGF-β ligands upon exposure to bacterial infection by testing double mutants. DAF-7 and TIG-3 are classified as TGF-β/Activin-like ligands, whereas DBL-1 and TIG-2 are BMP ligands. What is the rationale for the combinations used? For example, why TIG-3/DBL-1 was not studied?

4- The survival of dbl-1 and unc-129 animals show variability, which questions the reproducibility of the data and the reliability of the involvement of these ligands in defense (Fig 2 A, B).

5- Figure 3B lacks daf-7(m62) single animals. “Survival of daf-7;dbl-1 double mutants was not significantly different compared to dbl-1 single mutants (Line 228)”. However, there is difference between dbl-1(wa70) and daf-7(m62) animals in Fig 2A. Is there any difference between daf-7(m62) animals and double mutant animals?

6- How many animals were used per trial, and how many biological and technical replicates were used in the study?

Minor

Low contrast curves were used in survival figures Fig S3.

Lacks reference at line 217.

Line 237, a typo, Fig S3 rather Fig S1.

Fig S6 was mentioned ahead of Fig S4. Fig S5 was never mentioned.

Reviewer #3: In this manuscript, the authors examined the roles of five TGFbeta ligands in mediating the immune responses of C. elegans to two different gram-negative bacterial pathogens: Serratia marcescens or Photorhabdus luminescens. Using various single or double mutant combinations, they found that all five TGFbeta ligands contribute to some degree in the pathogen response, but not at the level of transcriptional induction. Furthermore, TIG-2/BMP and TIG-3/TGFbeta appear to exhibit non-additive roles in this process, and that they may function through the canonical BMP receptors and Smads. Finally, the authors provided evidence using computational modeling that TIG-2/BMP and TIG-3/TGFbeta have the potential of forming heterodimers in silico. The finding of a potential cross-subfamily heterodimerization of a BMP-like ligand (TIG-2) and a TGFbeta-like ligand (TIG-3) is quite novel, and will be of interest to the broad community of researchers working on TGFbeta signaling and immune responses.

I have several questions:

1) The authors showed that tig-2 dbl-1 double mutants exhibited a more severe phenotype than either single mutants in their response to P. luminescens, while tig-3; tig-2 double mutants behaved similarly to either tig-3 or tig-2 single mutants in their responses to P. luminescens. They concluded that TIG-2 and DBL-1 have additive, while TIG-2 and TIG-3 have cooperative roles, in the immune response. What are the phenotypes of tig-3; dbl-1 double mutants or tig-3; tig-2 dbl-1 triple mutants? Results from these analyses will help clarify the relationship among these three ligands in the immune response. The reason is the following: the authors first showed in Figure 1 that tig-2 dbl-1 double mutants exhibited a much more severe phenotype than either tig-2 or dbl-1 single mutants, but then showed in Figure 5 that sma-3 mutants behaved similarly to either tig-3 or tig-2 single mutants, but exhibited a more severe phenotype than dbl-1 single mutants. If as the authors argued that TIG-2 and TIG-3 function through the canonical BMP pathway mediated by SMA-3, does DBL-1 then function via a non-canonical BMP pathway?

2) It would be helpful to show in a supplemental Figure where the deletions are for the two tig-2 deletions alleles and how they may impact on the molecular structure of TIG-2.

3) It is unclear why there is no error bar for the control samples in the qRT-PCR results shown in Figure 2.

4) Please comment on why the control animals for sma-4 mutants exhibit a much shorter life span than other control animals in Figure 4.

Minor:

1) Line 77, Zamarron and Chen, 2011 (rather than 20011)

2) Line 430, “in the context of an intact organism” (rather than “the”)

3) Figure S3 legend: unc-129(ev554); tig-2(ok3416) (rather than unc-129(ev554);tig-2(3416)).

**Have all data underlying the figures and results presented in the manuscript been provided?**

Reviewer #1: Yes

Reviewer #2: Yes

Reviewer #3: Yes

PLOS authors have the option to publish the peer review history of their article (what does this mean?). If published, this will include your full peer review and any attached files.

Reviewer #1: No

Reviewer #2: No

Reviewer #3: No

---

## [Editor Report · Decision Letter 1]

28 May 2024

Dear Dr Savage-Dunn,

We are pleased to inform you that your manuscript entitled "TGF-β Ligand Cross-Subfamily Interactions in the Response of Caenorhabditis elegans to a Bacterial Pathogen" has been editorially accepted for publication in PLOS Genetics. Congratulations!

Yours sincerely,

Danielle A. Garsin

Academic Editor

PLOS Genetics

Gregory P. Copenhaver

Section Editor

PLOS Genetics

Comments from the reviewers (if applicable):

**Data Deposition**

http://datadryad.org/submit?journalID=pgenetics&manu=PGENETICS-D-23-01020R1

**Press Queries**

---

## [Editor Report · Acceptance letter]

7 Jun 2024

PGENETICS-D-23-01020R1 

TGF-β Ligand Cross-Subfamily Interactions in the Response of Caenorhabditis elegans  to a Bacterial Pathogen 

Dear Dr Savage-Dunn, 

We are pleased to inform you that your manuscript entitled "TGF-β Ligand Cross-Subfamily Interactions in the Response of Caenorhabditis elegans  to a Bacterial Pathogen" has been formally accepted for publication in PLOS Genetics! Your manuscript is now with our production department and you will be notified of the publication date in due course.

With kind regards,

Olena Szabo

PLOS Genetics

On behalf of:
